# A classical chiral spin liquid from chiral interactions on the pyrochlore lattice

Daniel Lozano-Gómez [1] ✉, Yasir Iqbal[2] & Matthias Vojta[1]

Classical spin liquids are paramagnetic phases that feature nontrivial patterns of spin correlations within their ground-state manifold whose degeneracy scales with system size. Often they harbor fractionalized excitations, and their low-energy fluctuations are described by emergent gauge theories. In this work, we discuss a model composed of chiral three-body spin interactions on the pyrochlore lattice that realizes a novel classical chiral spin liquid whose excitations are fractonalized while also displaying a fracton-like behavior. We demonstrate that the ground-state manifold of this spin liquid is given by a subset of the so-called color-ice states. We show that the low-energy states are captured by an effective gauge theory which possesses a divergence-free condition and an additional chiral term that constrains the total flux of the fields through a single tetrahedron. The divergence-free constraint on the gauge fields results in two-fold pinch points in the spin structure factor and the identification of bionic charges as excitations of the system.

Spin liquids are disordered yet highly correlated phases of matter whereby magnetic degrees of freedom evade symmetry-breaking long-range order down to lowest temperatures[1,2]. It has been shown how such cooperative behavior can be succinctly described by emergent gauge symmetries[3,4]. Frustrated Mott-insulating magnets have been established as the key platform to realize classical and quantum spin liquids, which can emerge as a consequence of competing interactions stemming from either the architecture of the underlying lattice or from strong spin-orbit coupling[5–12].

The pyrochlore lattice, composed of a network of corner-sharing tetrahedra with magnetic ions located at the vertices, has proven to be an excellent arena for the realization of spin liquid phases. In the classical realm, examples of such highly correlated phases include the well-known spin-ice phase[4,6,13,14], the recent realizations of rank-2 spin liquids[7–9,15–17] as well as mixed rank-1–rank-2 spin liquids[12]. All these liquid phases are realized in spin systems whose interactions are bilinear in the spin degrees of freedom, taking the form $\mathbf{S}_i \mathcal{H}_{ij} \mathbf{S}_j$. Here, the $\mathcal{H}_{ij}$ spin coupling matrix in the generic case encompasses both isotropic and anisotropic interactions between not only first but also farther neighbors. This includes the isotropic Heisenberg terms $(\mathbf{S}_i \cdot \mathbf{S}_j)$[5,18], as well as, anisotropic Ising $(S_i^z S_j^z)$[17], Dzyaloshinskii-Moriya

$(\mathbf{D}_{ij} \cdot [\mathbf{S}_i \times \mathbf{S}_j])$[19], and off-diagonal symmetric also known as pseudo-dipole $(S_i^x S_j^y + x \leftrightarrow y)$[20], as well as their analogs for further-neighbor interaction terms[21–23].

In contrast, much less attention has been devoted to spin Hamiltonians with three-body or four-body spin interactions, which might also offer the possibility of realizing spin liquid phases with other types of exotic emergent gauge symmetries at low temperatures. One example of such a higher-body interaction is the isotropic biquadratic interaction $(\mathbf{S}_i \cdot \mathbf{S}_j)^2$[24,25]. A recent work studied the physics resulting from the biquadratic coupling on the pyrochlore lattice with an additional Heisenberg term[24]. However, such a model features an order-by-disorder selection of a magnetically ordered state at low temperatures.

In the present paper, we consider the so-called scalar spin chiral term, a magnetic three-body interaction that arises in a $t/U$ expansion of the Hubbard model at half-filling in the presence of a magnetic field[26]. This leads to the following spin-rotation invariant Hamiltonian, which breaks time-reversal and parity symmetries[27,28]

$$\mathcal{H}_\chi = -J_\chi \sum_{i,j,k \in \Delta} \chi_{ijk} \,, \tag{1}$$

[1]Institut für Theoretische Physik and Würzburg-Dresden Cluster of Excellence ct.qmat, Technische Universität Dresden, 01062 Dresden, Germany.
[2]Department of Physics and Quantum Centre of Excellence for Diamond and Emergent Materials (QuCenDiEM), Indian Institute of Technology Madras, Chennai 600036, India. ✉e-mail: daniel.lozano-gomez@tu-dresden.de

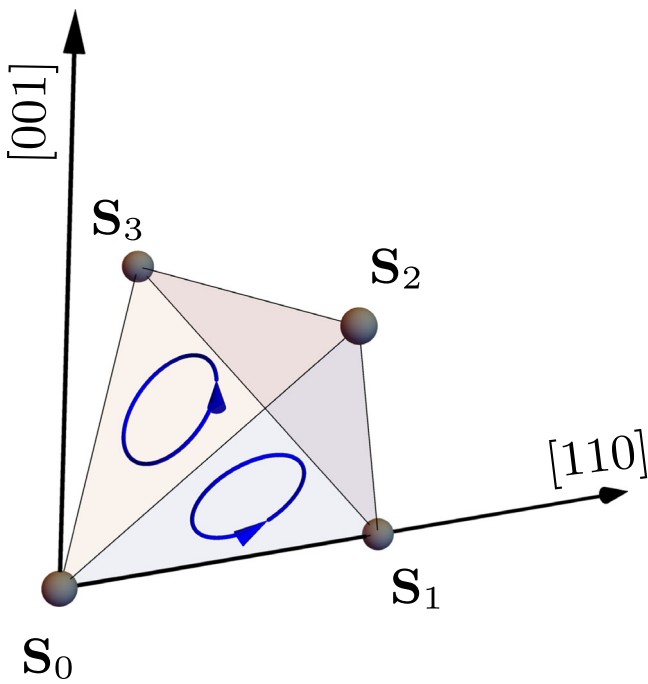

**Fig. 1 | Single-tetrahedron chirality.** Illustration of an up-tetrahedron with the direction of the chiral term specified for two of its faces, $\mathbf{S}_0 \cdot (\mathbf{S}_1 \times \mathbf{S}_2)$ and $\mathbf{S}_0 \cdot (\mathbf{S}_2 \times \mathbf{S}_3)$. Here, the black arrow denotes the high-symmetry [001] and [110] directions.

where $\chi_{ijk} = \mathbf{S}_i \cdot (\mathbf{S}_j \times \mathbf{S}_k)$, with $i, j, k$ being the corners of triangular faces in up and down-tetrahedra. We have chosen the chirality such that, for an up-tetrahedron, one term is $\mathbf{S}_0 \cdot (\mathbf{S}_1 \times \mathbf{S}_2)$, where $\mathbf{S}_0$ is located at [000], $\mathbf{S}_1$ at $\frac{1}{4}$[110], and $\mathbf{S}_2$ at $\frac{1}{4}$[101], see Fig. 1. Schematically, the chirality of every triangular face of a single tetrahedron is associated with a chiral vector pointing outwards of its corresponding tetrahedron, such that we have a uniform chiral model. The above Hamiltonian has previously been investigated on the Kagome lattice, where both classical and quantum order-by-disorder mechanisms drive the system into a long-range ordered state where the spins in every triangle are constrained to point along one of the three global Cartesian axis[29].

Here, we shall study the classical limit ($S \to \infty$) of the Hamiltonian in Eq. (1) on the pyrochlore lattice. We demonstrate that the system realizes a novel chiral classical spin liquid phase down to the lowest simulated temperatures. The ground-state manifold is characterized by the spins in every single tetrahedron pointing along four distinct directions, with a restriction stemming from the chirality of the Hamiltonian in Eq. (1). The identification of the constraints governing the ground-state manifold allows us to study this manifold through an effective 4-state Potts model on the pyrochlore lattice with an additional chiral term. The mapping to the 4-state Potts model permits us to identify three intertwined color gauge fields that fulfill an emergent Gauss' law and whose single-tetrahedron fluxes fulfill a right-hand rule in the ground-state manifold. The excitations of this emergent theory are comprised of confined bionic charges with restricted mobility originating from the energetically preferred right-hand rule between the intertwined color gauge fields. The properties of the elementary excitations, along with the thermodynamics of the system, suggest that this minimal model describes a fractonic system where the ground-state manifold is characterized by at least a sub-extensive degeneracy. This Hamiltonian on the pyrochlore lattice, therefore, constitutes a relatively simple spin model capable of realizing fracton physics, which might be instrumental in the study of fracton systems, the restricted mobility of its excitations, and its

intricate thermodynamics. The remainder of the paper is organized as follows: first, we start by considering the physics of an isolated tetrahedron subject to the chiral interaction, followed by classical Monte-Carlo results for the full lattice system. Then we discuss the low-energy manifold in terms of an effective Potts model and develop a corresponding gauge theory. Lastly, we present numerical results for a model which also includes nearest-neighbor Heisenberg interactions. A concluding section closes the paper, while technical details are relegated to the supplementary information.

## Results
### Single-tetrahedron analysis
As a first approach to the chiral Hamiltonian in Eq. (1), we study the single-tetrahedron case by numerically minimizing the energy of a four-spin configuration through an iterative minimization algorithm[12]. This minimization results in spin configurations where the dot product between two distinct spins equals (−1/3). This constraint is fulfilled by the spin orientations

$$\mathbf{u}_0 = \frac{1}{\sqrt{3}}(\bar{1}\bar{1}1), \qquad \mathbf{u}_1 = \frac{1}{\sqrt{3}}(11\bar{1}), \qquad (2)$$

$$\mathbf{u}_2 = \frac{1}{\sqrt{3}}(1\bar{1}1), \qquad \mathbf{u}_3 = \frac{1}{\sqrt{3}}(\bar{1}11), \qquad (3)$$

or equivalently by considering an all-out configuration of the spins, as shown in Fig. 2a. This constraint permits the construction of alternative ground-state configurations of a single tetrahedron, by applying an even permutation of the spin orientations in the single tetrahedron. Note that an odd permutation would instead result in a higher-energy configuration as a consequence of the triple-product term in the Hamiltonian in Eq. (1). To illustrate the construction of the aforementioned single-tetrahedron ground states, up to a global O(3) rotation, we associate the spin orientations {$\mathbf{u}_0$, $\mathbf{u}_1$, $\mathbf{u}_2$, $\mathbf{u}_3$} with a unique color, {Red, Blue, Green, Yellow} $\equiv$ {R, B, G, Y}, which we refer to as the coloring basis. This mapping identifies the all-out configuration illustrated in Fig. 2a with the colored configuration illustrated in Fig. 2b. For completeness, we note that there exists yet another representation of the single-tetrahedron configurations in terms of three emergent gauge fields $\mathbf{B}_\mu^{(c)}$ shown in Fig. 2c, which we discuss in detail in the subsequent sections. Using the coloring basis, we identify 12 distinct 4-color ground-state configurations obtained by applying even permutations on the all-out configuration; these are shown in Fig. 3.

We note that a spin configuration spanning the entire pyrochlore lattice can be constructed by assigning a spin configuration in Fig. 3 to all up tetrahedra while keeping the down-tetrahedra in an equivalent low-energy configuration. These configurations possess an energy of $E_0 = -1.5396 J_\chi$ per lattice site.

### Numerical simulations
To investigate the ground states and thermodynamics of the Hamiltonian in Eq. (1) on the full pyrochlore lattice, we perform classical Monte-Carlo (cMC) and iterative minimization (IM) simulations considering systems comprised of $4L^3$ spins with systems of size $L = 10$. To thermalize our system, we implement a Gaussian single-spin-flip update[30], an over-relaxation algorithm[31–33], and a multi-valley average between independent cMC simulations inspired by the study of spin glasses[34]. In addition to thermodynamic quantities, we compute the equal-time spin-structure factor.

Figure 4 shows the internal energy and specific heat of the system obtained from a cooling scheme. These quantities smoothly evolve down to low $T$, with the specific heat plateauing at a value of $C/k_B = 1$ and the energy per site tending to $E \sim -1.52 J_\chi$. No signatures of a transition to a symmetry-broken phase are visible. We note that, although

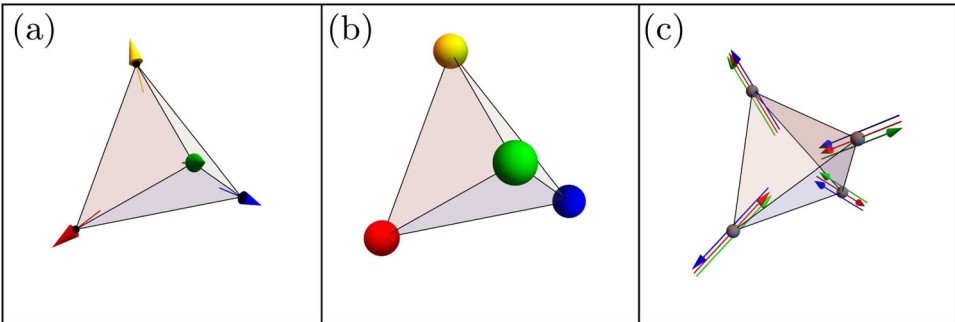

**Fig. 2 | Single-tetrahedron spin, Potts, and gauge-field configuration.** Example of a single-tetrahedron minimum-energy configuration for the chiral Hamiltonian in Eq. (1) shown in the Heisenberg spin configuration (**a**), the color representation (**b**), the Potts gauge fields $\mathbf{B}_\mu^{(x)}$, $\mathbf{B}_\mu^{(y)}$, and $\mathbf{B}_\mu^{(z)}$ colored in red, blue, and green, respectively, in the single tetrahedron (**c**).

**Fig. 3 | Allowed color configurations.** Single-tetrahedron ground states $S^\chi$ of the chiral Hamiltonian in Eq. (1) in the color basis where the colors red, blue, green, and yellow correspond to the spin orientations $\mathbf{u}_0$, $\mathbf{u}_1$, $\mathbf{u}_2$, and $\mathbf{u}_3$ as defined in the main text, respectively.

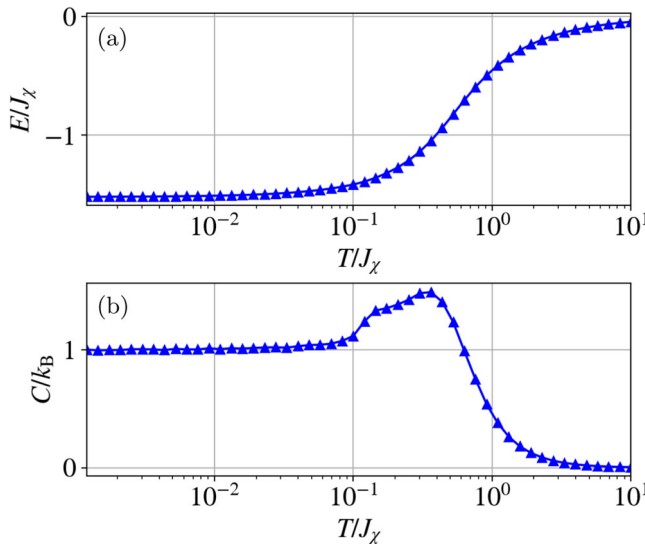

**Fig. 4 | Classical Monte-Carlo thermodynamics. a** Internal energy and **b** specific heat per site of the spin chiral Hamiltonian in Eq. (1), showing a smooth evolution as a function of temperature. Note that the specific heat develops a double bump at temperatures of order $10^{-1}J_\chi$.

the measured internal energy of the system is very close to the value obtained from the single-tetrahedron analysis, there is a small deviation when this is measured in a cooling-down scheme in cMC. We discuss this deviation, as well as the double-bump structure in $C(T)$, in the next section.

To further investigate the finite-temperature behavior of the chiral Hamiltonian in Eq. (1), we study the temperature evolution of the (equal-time) spin-structure factor in three distinct temperature regimes: one for temperatures above the double-bump features, one chosen within the temperature window comprising the double bump, and a temperature well below these features where we use both MC and IM to improve the statistics of the measurement (we refer the reader to the Supplementary Information (SI) for the precise expressions of the correlation functions). The resulting structure factor is shown in the $[hh\ell]$ and $[hk0]$ planes in Fig. 5 for the three temperatures considered. At high temperatures, broad features are observed indicating an uncorrelated paramagnetic regime, see Fig. 5a. As the temperature is decreased, these features sharpen up and lead to two-fold pinch points[4,9–11,35–39], see Fig. 5b, resulting in connected bow-tie and diamond patterns in the $[hh\ell]$ and $[hk0]$ planes, respectively. The two-fold pinch points reflect dipolar correlations between the spin degrees of freedom and are indicative of an energetically imposed Gauss' law constraint on certain gauge fields $\mathbf{B}^{(c)}$, namely $\nabla \cdot \mathbf{B}^{(c)} = 0$, describing an effective low-temperature theory of the system[5,39]. In particular, within the double-bump temperature window, additional features in the diamonds and bow-tie patterns appear. These anisotropic features become more pronounced as the temperature is lowered below the double-bump window resulting in a cross-like pattern in the $[hk0]$ plane and a dip along the direction of the two-fold pinch points in the $[hh\ell]$ plane, see Fig. 5c.

At high temperatures, the structure factor profile (the two-fold pinch points and its location) is qualitatively similar to that observed for the pure Heisenberg antiferromagnetic model (HAFM). In the HAFM, the two-fold pinch points are associated with an emergent gauge field abiding by a Gauss' law, which in terms of the spin configurations, translates into a vanishing magnetization in every tetrahedron[5]. Therefore, the observation of these features in the present model suggests that a similar vanishing magnetization constraint might be present. Indeed, a study of the magnetization distribution reveals that the system realizes a vanishing single-tetrahedron

magnetization as the temperature decreases, see SI for more details. This indicates the observation of an energetic antiferromagnetic constraint governing the low-temperature configurations.

The vanishing single-tetrahedron magnetization and the two-fold pinch-point features observed in the spin-structure factors suggest that the ground-state manifold of the Hamiltonian in Eq. (1) is conformed by a variety of antiferromagnetic configurations. However, the presence of additional features in the structure factor suggests that further constraints, in addition to the vanishing single-tetrahedron magnetization, exist in the ground-state manifold. This observation is in line with the single-tetrahedron analysis whose spin configurations, i.e., those shown in Fig. 3, are antiferromagnetic while the spins in a single tetrahedron are constrained to point along the directions $\{\mathbf{u}_0, \mathbf{u}_1, \mathbf{u}_2, \mathbf{u}_3\}$, up to a global O(3) rotation.

To study the onset of this additional constraint as a function of temperature, we measure the distribution of the dot product between nearest-neighbor spins. As seen in Fig. 6a, the distribution develops a peak at the value (−1/3) for temperatures below the double-bump feature in the specific heat, while remaining relatively featureless for higher temperatures. As the temperature is further decreased below the double-bump feature, the distribution becomes sharper while remaining centered at the value of (−1/3), suggesting that in the $T \to 0$ limit, the ground-state configurations are those predicted by the single-tetrahedron analysis. Consequently, we associate the onset of this peak in the distribution with the system entering a temperature regime where the spins in the system progressively adopt a colored configuration.

On passing, we note that for the Hamiltonian in Eq. (1) on the Kagome lattice[29] a similar double-bump feature in the specific heat was also observed and associated with the system entering a temperature regime where the spins in a triangle are confined to be pointing along one of the global Cartesian directions[29]. In such a case, however, the low-$T$ specific heat reaches $C/k_B = 11/12$, a value associated with quartic spin fluctuations above the ground-state configuration leading to the entropic selection of a symmetry-breaking configuration at low temperatures over a finite (due to Mermin-Wagner theorem) yet progressively growing correlation radius.

## Thermalization and freezing

The lowest energy measured in a cool-down cMC scheme is $E(T \to 0^+) \simeq -1.52J_\chi$ (per site), close to the single-tetrahedron ground-state energy of $E_0 = -1.5396J_\chi$. However, it is important to note that in a cool-down scheme, our cMC simulations seem to plateau at an energy slightly higher than $E_0$. To address the origin of this discrepancy, we consider a warm-up scheme starting from an all-out configuration at $T = 0^+$ and compare the evolution of the dot-product distribution and the internal energy with those obtained from a cool-down scheme, see Figs. 6b and Fig. 7. As observed for the cool-down scheme, the distribution of the dot product of the warm-up scheme develops a peak centered at (−1/3) at low temperatures while appearing to be featureless at high temperatures. However, we note that at low temperatures, the distribution of the warm-up scheme appears to be sharper compared to that obtained from a cool-down scheme at the same temperatures.

The discrepancy between the cool-down and warm-up evolution procedures can also be observed in the internal energy of the system. Indeed, the energies obtained at low $T$ within the warm-up scheme are consistent with the single-tetrahedron ground-state energy $E_0$, whereas the cool-down scheme levels off at a higher value, see Fig. 7. Although the warm-up scheme better represents the expected internal energy, we note that this procedure appears to be "frozen" in the initial all-out state up to temperatures where the double-bump structure in Fig. 4 is observed. Indeed, the specific heat from the warm-up scheme shows a distinct peak associated with a crossover from a low-temperature ordered phase, the all-out order, to a high-temperature

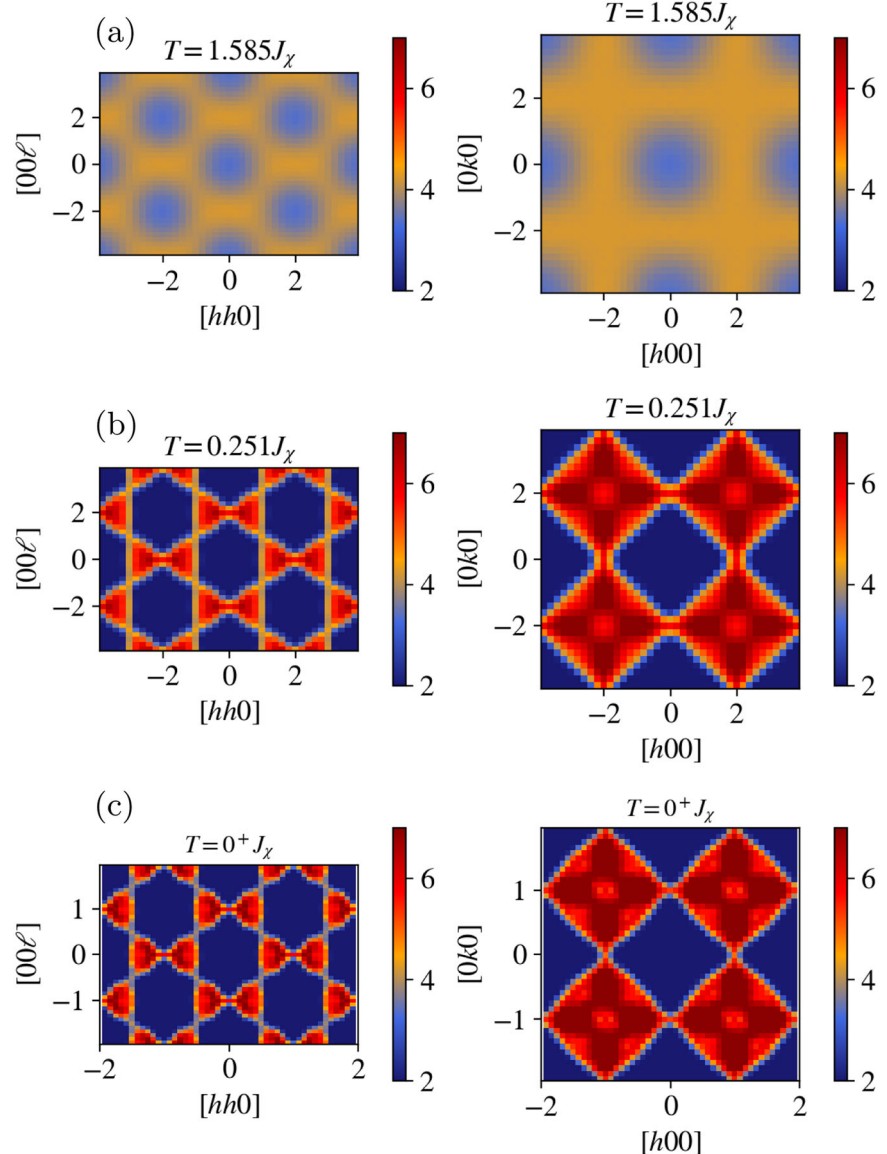

**Fig. 5 | Spin correlation functions.** Equal-time spin-structure factor for three distinct temperatures in the high-symmetry [$hh\ell$] (left column) and [$hk0$] (right column) scattering planes for the Hamiltonian in Eq. (1) where the formation of sharp two-fold pinch points is observed as the temperature is decreased. Here, panels **a**, **b**, and **c** correspond to structure factors in the [$hh\ell$] (left) and [$hk0$] (right) scattering planes sampled at $T = 1.585 J_\chi$, $T = 0.251 J_\chi$, and in the $T \to 0^+ J_\chi$ limit, respectively.

disordered phase. The discrepancy between these two schemes, in addition to the freezing of the warm-up scheme, suggests that the cool-down scheme finds a locally stable configuration while the warm-up scheme is trapped in the global energy minimum. Indeed, we have verified that this low-temperature freezing is also observed when the starting warm-up state is not a perfect **k** = **0** state, suggesting that the freezing at low temperatures is independent of the starting 4-color configuration for reasons we discuss below. For more details on the warm-up scheme, we refer the reader to the SI.

The variation of thermodynamic quantities measured depending on the different sampling schemes is characteristic of spin-glass systems[40–42], of certain spin liquids where non-local updates are needed to tunnel between distinct ground-state configurations or to move and annihilate excitations[12,16], and of fractonic systems[43,44]. In the next section, we discuss such a scenario by identifying an effective gauge theory describing the ground-state manifold, which reveals the emergence of complex gauge charges that are directly correlated with the freezing and responsible for the mismatch between the warm-up

and cool-down schemes. For more details on the evolution of the cMC results and the cool-down procedure, we refer the reader to the SI.

## Effective Potts model and ground-state manifold

The construction of the ground-state manifold is greatly simplified by considering distinct tiling patterns of 4-color states, however, this construction does not provide us with an effective theory describing the low-temperature physics of this model. Indeed, a common hallmark of classical spin liquids is the emergence of a low-energy field theory that associates the constraints of the ground-state manifold with the appearance of a gauge symmetry[45].

Nevertheless, the characterization of the ground-state manifold employing the 4-color mapping suggests that a theory describing the low-temperature spin liquid phase is associated with a type of antiferromagnetic $q$-state Potts Hamiltonian with $q = 4$ whose ground-state manifold is given by the 4-color states shown in Fig. 3. A similar mapping into an effective Potts model at low temperatures was performed in refs. 46,47; this mapping was crucial when exposing the

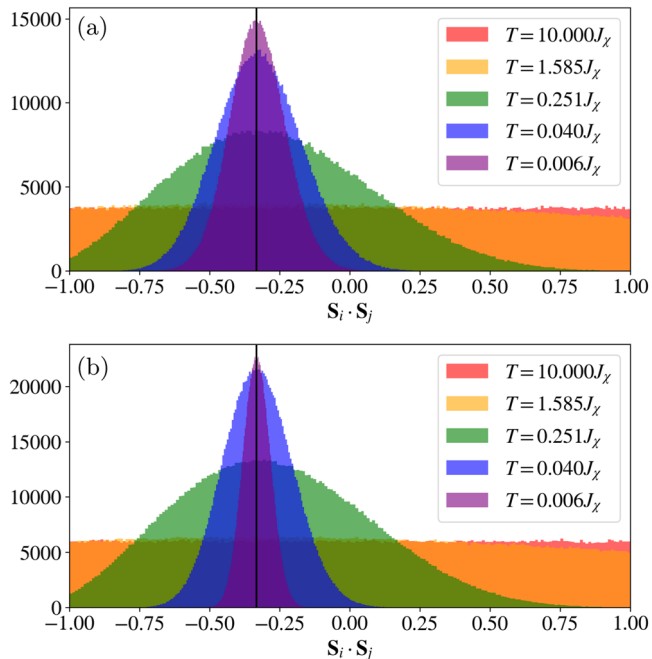

**Fig. 6 | Cool-down and warm-up nearest-neighbor histograms.** Histogram of the nearest-neighbor spin correlation, $\mathbf{S}_i \cdot \mathbf{S}_j$, for distinct configurations sampled from our cMC simulations for different temperatures obtained for **a** a cool-down and **b** a warm-up scheme. The vertical lines mark the value (−1/3), the predicted dot product between neighboring spins in the ground-state manifold.

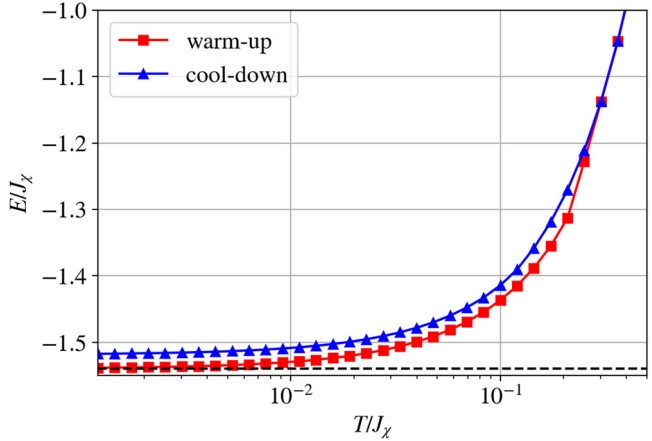

**Fig. 7 | Warm-up and cool-down classical Monte-Carlo comparison.** Internal energy per lattice site was obtained using a warm-up and a cool-down scheme in a classical Monte-Carlo simulation. The dashed line indicates the energy $E_0 = -1.5396J_\chi$ of the single-tetrahedron ground state.

ground-state constraints governing the low-energy manifold. The regular 4-state Potts model, however, allows all 4-color states per tetrahedron. Thus, in order to preserve only the configurations listed in Fig. 3, we consider the modified Potts Hamiltonian

$$\mathcal{H}_\chi^{\text{Potts}} = J \sum_{\langle i,j \rangle} \delta_{q_i, q_j} + J_\chi \sum_{\boxtimes} \left(1 - \delta\left(S^{\boxtimes}, S^\chi\right)\right), \tag{4}$$

where the first term corresponds to the usual Potts interaction, and the second term corresponds to an energy cost $J_\chi$ given a 4-color state $S^{\boxtimes}$ that is not a part of the list of 4-color states $S^\chi$ shown in Fig. 3, i.e., the configurations obtained by performing odd permutations on the all-

out 4-color state. We refer to this second term as a chiral term for reasons which will become clear in the subsequent discussion.

By construction, the ground-state configuration of the Hamiltonian in Eq. (4) (with the built-in constraint of having vanishing internal energy) corresponds to ground-state configurations of the chiral Hamiltonian in Eq. (1). Indeed, such equivalence can be numerically established by obtaining ground-state configurations from the Hamiltonian in Eq. (4) via a cMC and then translating these into the corresponding Heisenberg spin configurations to successively compute the energy for the Hamiltonian in Eq. (1). We note that this equivalence can also be tested analytically, as the energy of all configurations in Fig. 3 is replicated in all tetrahedra assuming a perfect tiling can be performed in the full lattice. For more details on the chiral Potts mode in Eq. (1), we refer the reader to the SI.

## Gauge structure of the regular Potts model

Having demonstrated that the two models in Eqs. (1) and (4) result in a similar ground-state manifold, we now construct an effective gauge theory capable of describing the correlations observed in the ground-state manifold of the Potts Hamiltonian and, by extension, the chiral Hamiltonian. As a starting point, we consider the regular Potts model, i.e., the one with $J_\chi = 0$, whose gauge theory on the pyrochlore lattice for the antiferromagnetic case was studied in Ref. [48]. The regular 4-state Potts model in this lattice can be described by an effective field theory where three intertwined gauge fields $\{\mathbf{B}_\mu^{(c)}\}$ identified by the index $c \in \{x, y, z\}$ are defined as

$$\mathbf{B}_\mu^{(c)}(\mathbf{r}) = S_\mu^c(\mathbf{r})\mathbf{z}_\mu, \tag{5}$$

where $\mu$ labels the sublattice index, $\mathbf{r}$ denotes an FCC lattice vector, $c$ also indexes the spin component, and $\mathbf{z}_\mu$ is the local $z$-direction of the spin in sublattice $\mu$, see the SI for the definition of the local $z$ directions. On this basis, the all-out configuration is associated with the gauge-field configuration illustrated in Fig. 2c. In the ground-state manifold, the spin $\mathbf{S}_\mu(\mathbf{r})$ corresponds to one of the four possible color orientations $\{R, B, G, Y\} \equiv \{\mathbf{u}_0, \mathbf{u}_1, \mathbf{u}_2, \mathbf{u}_3\}$. At low temperatures, the three intertwined fields follow an energetically imposed 2-In-2-Out constraint indicating an emergent Gauss' law $\nabla \cdot \mathbf{B}^{(c)} = 0$, see SI for all the ground-state single-tetrahedron gauge-field configurations. This construction identifies an effective Hamiltonian for the $q = 4$ Potts model provided by

$$\mathcal{H}_{\text{eff}}(J_\chi = 0) \propto \int d\mathbf{r} \left[ J \sum_c |\nabla \cdot \mathbf{B}^{(c)}(\mathbf{r})|^2 \right]. \tag{6}$$

This effective low-temperature gauge-field theory in Eq. (6) implies that in the ground-state manifold the gauge fields fulfill a divergence-free condition[4,49], equivalent to that of spin-ice, indicating that the field lines associated with these fields can have no boundaries and therefore consist of closed loops.

In Fig. 8a, we show a gauge-field configuration for a state in the ground-state manifold. Extending the similarities with the spin-ice phase, distinct ground-state configurations of this model can be obtained by identifying closed loops conformed by two colors and then interchanging the colors in the loop. In contrast with spin-ice, violations of the divergence-free condition result in the generation of two gauge charges, dubbed bions, which violate the divergence-free constraints of two gauge fields concurrently and result in an energetic cost proportional to $J$. For the regular Potts Hamiltonian, the bions are free to move with no additional energy cost and are connected by "Dirac strings" colored by the gauge fields associated with the bions. In Fig. 8b, we illustrate a high-energy gauge configuration resulting from applying an even permutation of the color degrees of freedom to the tetrahedron in the center of Fig. 8a. This permutation results in the generation of 8 bionic charges where the divergence-free constraint is

(a)　　　　　　　　　　　　　(b)

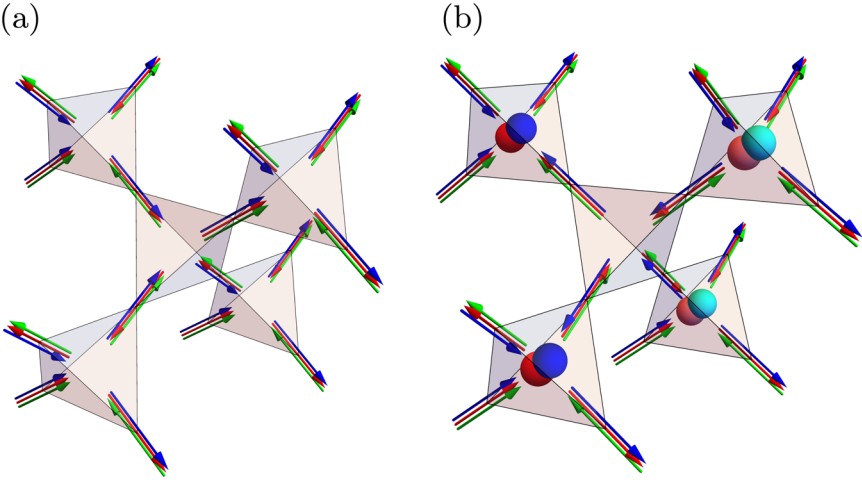

**Fig. 8 | Ground state and excited state gauge-field configuration.** Configuration of the Potts gauge fields $\mathbf{B}_\mu^{(c)}$ for **a** a ground-state $\mathbf{k} = \mathbf{0}$ configuration, and **b** an excited configuration where bionic excitations are present. Here, the red, blue, and green arrows denote the local orientations of the $\mathbf{B}_\mu^{(x)}$, $\mathbf{B}_\mu^{(y)}$, and $\mathbf{B}_\mu^{(z)}$ fields, respectively. Note that, for the ground-state configuration, every tetrahedron obeys a two-in–two-out rule for each Potts field, corresponding to a state with no charges associated with any field. On the other hand, for the excited configuration, the red (blue) spheres represent charges associated with a non-zero Gauss' law of the $\mathbf{B}_\mu^{(x)}$ ($\mathbf{B}_\mu^{(y)}$) field in the corresponding tetrahedra, where the light (dark) color indicates that the charge is positive (negative).

broken in the tetrahedra where the light-colored (dark-colored) bions correspond to positively (negatively) charged bions of the corresponding color field.

## Gauge structure of the chiral Potts model

Let us now consider the full Potts Hamiltonian with $J_\chi > 0$ in Eq. (4), which restricts the configurations to only those shown in Fig. 3. It is clear that the inclusion of this chiral term does not change the emergent Gauss' law in the $\mathbf{B}_\mu^c$ fields as the subset of configurations allowed by the chiral term still fulfills the constraint of having a vanishing $\nabla \cdot \mathbf{B}^{(c)}$. The introduction of this term, however restricts the allowed chirality between the total flux of the three gauge fields in a single tetrahedron, defined as

$$\mathbf{\Phi}^{(c)}(\mathbf{r}) \equiv \sum_{\mu=0}^{3} \mathbf{B}_\mu^{(c)} = \sum_{\mu=0}^{3} S_\mu^c(\mathbf{r})\mathbf{z}_\mu, \tag{7}$$

which, in the ground-state manifold, are constrained to be mutually perpendicular, see the SI. In other words, the introduction of the chiral term only permits those color configurations for which the product

$$\mathbf{\Phi}^{(x)} \cdot (\mathbf{\Phi}^{(y)} \times \mathbf{\Phi}^{(z)}), \tag{8}$$

yields a positive value.

To make this observation mathematically precise it suffices to associate the even and odd permutations of the permutation group $\mathcal{S}_4$ with the proper and improper rotations of the tetrahedral group $T_d$, respectively. This separation of both the $\mathcal{S}_4$ and the $T_d$ groups can be performed by considering the equivalence classes associated by the sign of the permutation and the determinant of the transformation, respectively. The equivalence classes then allow us to identify every even (odd) permutation of the group $\mathcal{S}_4$ with a proper (improper) rotation in $T_d$. Indeed, note that the single-tetrahedron configurations shown in Fig. 3 can be obtained from proper rotations of the single-tetrahedron group $T_d$ starting from the all-out configuration in Fig. 2b. By definition, proper rotations do not change the chirality of a mathematical construct, whereas improper rotations do. This property implies that the chirality between the fluxes in the ground-state manifold is a "built-in" energetic restriction of the Hamiltonian considered in Eq. (4). Consequently, and motivated by the effective theory of the regular Potts model, we propose a minimal phenomenological

effective gauge theory that encompasses all the restrictions for the gauge fields

$$\mathcal{H}_{\text{eff}} = \int d\mathbf{r} \left[ J \sum_c |\nabla \cdot \mathbf{B}^{(c)}|^2 - J_\chi \mathbf{\Phi}^{(x)} \cdot (\mathbf{\Phi}^{(y)} \times \mathbf{\Phi}^{(z)}) \right], \tag{9}$$

where the first term constrains the gauge fields to be divergence-free while the second term enforces the right-hand chirality between the fluxes, with $J_\chi$ being defined as a phenomenological positive constant. Note that the proposed theory breaks time-reversal symmetry as it is composed of both two-body and three-body terms. This is natural given that the original Hamiltonian in Eq. (1) breaks time-reversal symmetry as well.

## Excitations of the chiral Potts model

The introduction of the chiral term in the Hamiltonian adds an energy cost $J_\chi$ to the divergence-free gauge-field configurations, which results in a left-hand chirality of the total fluxes $\mathbf{\Phi}^{(c)}$. This apparent small modification to the Potts Hamiltonian results in crucial differences in the ground-state manifold of the Potts model and its excitations, now being bions and left-hand chiral fields. Indeed, flipping closed colored loops connecting two different ground-state configurations of the regular Potts model now results in high-energy configurations of the chiral Potts model whose energy grows proportional to the length of the closed loop: closed loops can be regarded as a chain of odd single-tetrahedra permutations where each permutation has an energy cost of $J_\chi$, i.e. for each left-hand chiral tetrahedra resulting from this permutation there is associated energy cost proportional to $J_\chi$.

Consequently, the Dirac strings connecting the bionic excitations now have a tension, associated with its length, resulting in the confinement of the bionic charges. It is then natural to ask what type of a non-local update connects distinct ground-state configurations that are constructed from even permutations in all the tetrahedra involved. Note, however, that the relation of this update with proper rotations (even permutations) implies that such a non-local transformation must be a closed 2-dimensional surface as all tetrahedra involve at least three of its four corners. Similar types of transformations have been studied in fractonic systems where the closed surfaces can be associated with the creation and posterior annihilation of fractonic charges[43].

Indeed, starting from a $\mathbf{k} = \mathbf{0}$ state, the simplest non-local transformation is identified as one that produces an even permutation in all the triangles of an aleatory selected Kagome plane. Through this sole transformation, we can identify a minimum degeneracy in the ground-state manifold, which scales with the linear size $L$, and not with the number of sites $L^3$, indicating that the ground-state manifold has at least a sub-extensive degeneracy associated with these transformations. These higher-dimensional non-local updates can be associated with the restricted motion of the bionic charges, suggesting that the bionic charges of the regular Potts model are fracton charges in the chiral Potts model.

The restricted motion of these excitations results in glassy dynamics typically observed in fracton systems[43]. Indeed, performing a warm-up and cool-down classical Monte-Carlo simulation on the chiral Potts model exposes a similar behavior to the one observed in Fig. 7: in the cool-down scheme, the restricted motion results in freezing of charges at low temperatures yielding an internal energy significantly above the ground-state energy. In contrast, in the warm-up scheme excitation cannot easily be created, and the system freezes in the initial configuration. We refer the reader to the SI for further details on the chiral Potts model. Similarly, for the chiral model in Eq. (1), the fractonic nature of these charges is responsible for the disagreement between the internal energy illustrated in Fig. 7, as well as the histograms of the nearest-neighbor spin correlation in Fig. 6 measured in the warm-up and cool-down schemes. In the chiral Hamiltonian, however, the continuous nature of the spin degrees of freedom allows for a slow thermal depopulation of the gauge charges and, therefore, a decreasing internal energy with decreasing temperature. The presence of these charges, however, is reflected in the width of the histograms in Fig. 6, which are consistently broader in the cool-down simulations. For more details regarding the difference between the cool-down and warm-up schemes for the chiral Hamiltonian in Eq. (1), we refer the reader to the SI.

## Chiral and Heisenberg interactions

As previously mentioned, the chiral Hamiltonian in Eq. (1) is one of the lower-order corrections when considering a Hubbard model with an applied magnetic field. This chiral term, however, is obtained as the next-to-leading order interaction after the usual Heisenberg interaction. It is, therefore, natural to inquire about the behavior of the Hamiltonian

$$\mathcal{H} = J \sum_{\langle ij \rangle} \mathbf{S}_i \cdot \mathbf{S}_j - J_\chi \sum_{i,j,k \in \Delta} \chi_{ijk}, \tag{10}$$

which now includes both the chiral and the Heisenberg term $J$ with $J > 0$. Naïvely, one could expect that the introduction of the Heisenberg interaction radically changes the overall physical behavior of the system depending on the ratio of the interactions $J/J_\chi$. However, as was previously discussed from the cMC simulations, the ground-state manifold resulting from the chiral interaction obeys a vanishing magnetization constraint which is the sole constraint imposed by the Heisenberg interaction. Consequently, the ground-state manifold of the Hamiltonian in Eq. (10) matches that of the Hamiltonian in Eq. (1). However, this does not imply that the thermodynamics of these systems are equivalent. Indeed, the introduction of a Heisenberg interaction modifies the behavior of the specific heat whereby the double-bump structure observed for the chiral Hamiltonian (1) is no longer present for a sufficiently large value of $J$ and is instead replaced by a single broad bump, see Fig. 9.

The location in temperature of this single bump appears to be relatively stable to the value of $J$, suggesting a relation of this feature with the chiral interaction parameter $J_\chi$ and the onset of the chiral constraints on the ground state. Indeed, similar to the case for pure

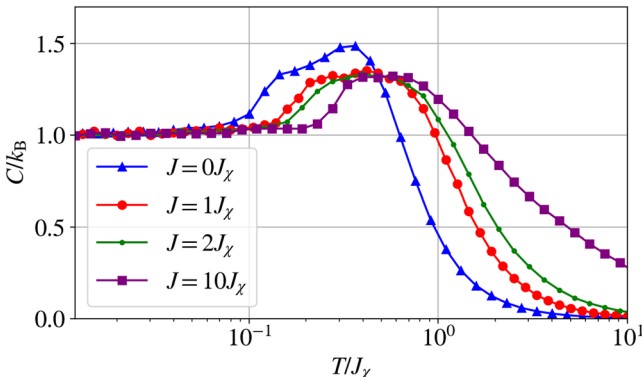

**Fig. 9 | Effect of an additional Heisenberg exchange interaction on the specific heat.** Specific heat of the Hamiltonian in Eq. (10) for various values of the Heisenberg coupling $J$. Here, the double-bump feature is only seen for the case $J = 0$ and $J = J_\chi$, whereas a single bump is observed for higher values of $J$.

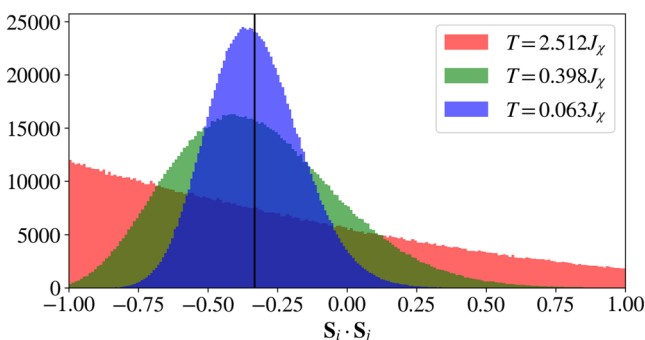

**Fig. 10 | Effect of an additional Heisenberg exchange interaction on the nearest-neighbor correlations.** Histogram of nearest-neighbor spin correlations for three different temperatures measured for the Hamiltonian in Eq. (10) with $J = 10J_\chi$.

chiral Hamiltonian for which $J = 0$, by studying the distribution of the dot product between neighboring spins, we find that this distribution develops a peak close to the value of $(-1/3)$ when this bump is reached and keeps on sharpening and approaching this value as the temperature is further decreased, see Fig. 10 for the case of $J = 10J_\chi$. It is worth mentioning that the shape of this distribution in the antiferromagnetic Heisenberg Hamiltonian with no chiral interaction, i.e., $J_\chi = 0$ and $J = 1$, resembles the shape of the high-temperature distribution in Fig. 10 down to the lowest temperature, see SI for further details on the evolution of this distribution.

## Discussion

We have studied the classical limit of the chiral Hamiltonian in Eq. (1) on the pyrochlore lattice using numerical and analytical tools to describe its thermodynamics and characterize the Hamiltonian's ground-state manifold. Our results suggest that the chiral Hamiltonian realizes a classical spin liquid phase at low temperatures where the excitations behave as fractons, i.e., quasiparticles with restricted mobility[36,37,43,50]. We model the ground-state manifold of this novel chiral spin liquid by identifying 4 distinct orientations where the spins in a single tetrahedron are constrained to point along (up to global O(3) rotations) in the $T \to 0$ limit. This constraint allows us to characterize the ground-state manifold in terms of a 4-state Potts model[48] and identify an emergent gauge theory. The effective theory employs three rank-1 fields which, in the ground-state manifold, become divergence-free and whose intertwined fluxes in a single tetrahedron follow a right-hand rule. The constraints found in the

ground-state manifold lead to a sub-extensive degeneracy which can be directly associated with even permutations of the spin states in all the triangles of an infinite Kagome plane bisecting the system in two.

The emergent gauge theory identifies the elementary excitations of the system as so-called bions, previously identified as the deconfined elementary excitation of a regular 4-state Potts model[48], acquire a restricted motion associated with the right-hand rule imposed in the total fluxes through a single tetrahedron in the chiral model. To further investigate the thermodynamics of its elementary excitations and its restricted motion, a non-local numerical algorithm tailored to this system must be developed[51,52]. The elementary excitations, the sub-extensive degeneracy, and the classical Monte-Carlo simulations presented are all consistent with the typical behavior observed in fracton systems. We emphasize that the restriction on the chirality of the gauge fluxes and the restricted motion of the associated gauge charges fundamentally differentiate the chiral spin liquid realized for the model in Eqs. (1) and (10) from previously identified pyrochlore spin liquid phases. Additionally, and of particular interest for the study of fracton models, the realization of fractonic charges identifies the chiral Hamiltonian in Eq. (1) on the pyrochlore lattice as a "simple" fracton model whose further study may shed light on the intricate physics associated with these systems.

Finally, we also considered the extension of the chiral model by an additional antiferromagnetic Heisenberg coupling. We demonstrated that the overall properties of the model remain largely unchanged for all strengths of the Heisenberg antiferromagnetic couplings considered.

As we have previously discussed, the chiral interaction in Eq. (1) descends from a $t/U$ expansion in the presence of a magnetic field[26]. To realize the sign structure of Eq. (1) would require a local magnetic field pointing towards (away from) the center of each tetrahedron. Such a local magnetic field is known to be realized internally in pyrochlore iridates $A_2Ir_2O_7$, where the A ions are typically rare-earth elements, and both the A and the Ir ions occupy two interpenetrating pyrochlore lattices. In this family of compounds, the Ir ions may undergo a phase transition into an all-in-all-out symmetry-breaking phase at a temperature well above the strength of the exchange interaction of the rare-earth ions[53]. The magnetic order in the Ir ions then results in an effective molecular field along the local $z$-direction for the rare-earth ions on the A sites[53]. Following Ref. 26, the introduction of such a weak local magnetic field may result in the chiral spin interactions we considered. Consequently, we identify the family $A_2Ir_2O_7$ of compounds where the Ir ions order into an all-in-all-out symmetry-breaking phase to be the natural candidates for the realization of the chiral interaction and, therefore, the chiral spin liquid we have introduced and studied in the present work.

Furthermore, we note that even if the Hamiltonian for a candidate material in this family of compounds does not exactly match the interaction couplings considered in Eq. (10), the sole proximity to this spin liquid may yield remnant thermodynamic features associated with the spin liquid. Indeed, such remnant features behavior has been predicted and observed for other spin liquids in the pyrochlore lattice, this being the case for $Yb_2Ti_2O_7$[54] and $FeF_3$[55], and even for other frustrated lattices as is the case for the recently synthesized trillium lattice compounds $K_2Ni_2(SO_4)_3$[56].

A natural extension of this work would be to consider a model with a staggered pattern of chirality on up- and down-tetrahedra, similar to what has been studied on the Kagome lattice[29]. Moreover, in the context of material realizations, it would be worthwhile to assess the fate of the spin liquid phase upon adding anisotropic couplings. Last but not least, studying the quantum counterpart of the present chiral model could yield a yet unexplored chiral fractonic quantum spin liquid.

## Methods

### Monte-Carlo simulations

Classical Monte-Carlo (cMC) simulations were performed on systems of size $L \in \{10, 12\}$, corresponding to $N = 4L^3$ classical spins with $|\mathbf{S}_i| = 1$, where we used $4 \times 10^4$ thermalization sweeps and $8 \times 10^4$ measurement sweeps. For each sweep, the system was updated using two update algorithms: a Gaussian update[30], and a over-relaxation[31]. Additionally, we performed an average of 10 to 100 independent MC simulations. We also implemented a cMC 4-state Potts cMC with a single-spin-flip update where the color of a site is randomly proposed and accepted with the usual Boltzmann weight.

## Data availability

This is a theoretical work with no experimental data produced. The processed Monte-Carlo data are available in the publicly available repository https://github.com/daniel-lozano/Classical_chiral_spin_liquid. Further numerical data that support the findings of this study are available from the authors upon request.

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

## Acknowledgements

We thank Kai Chung, Pedro Consoli, and Han Yan for their helpful discussions. D.L.-G and MV acknowledge financial support from the DFG through the Würzburg-Dresden Cluster of Excellence on Complexity and Topology in Quantum Matter—ct.qmat (EXC 2147, project-id 390858490) and through SFB 1143 (project-id 247310070). D.L.-G. is supported by the Hallwachs-Röntgen Postdoc Program of ct.qmat. The work of Y.I. was performed, in part, at the Aspen Center for Physics, which is supported by National Science Foundation Grant No. PHY-2210452. The participation of Y.I. at the Aspen Center for Physics was supported by the Simons Foundation (1161654, Troyer). The research of Y.I. was carried out, in part, at the Kavli Institute for Theoretical Physics in Santa Barbara during the "A New Spin on Quantum Magnets" program in summer 2023, supported by the National Science Foundation under Grant No. NSF PHY-2309135. Y.I. acknowledges support from the ICTP through the Associates Program and from the Simons Foundation through Grant No. 284558FY19, IIT Madras through the Institute of Eminence (IoE) program for establishing QuCenDiEM (Project No. SP22231244CPETWOQCDHOC), and the International Center for Theoretical Sciences (ICTS), Bengaluru, India during a visit for participating in the program "Frustrated Metals and Insulators" (Code No. ICTS/frumi2022/9). Y.I. also acknowledges the use of the computing resources at HPCE, IIT Madras.

## Author contributions

Y.I. formulated the project. D.L.-G. performed the analysis and interpretation of the single-tetrahedron analysis and Monte-Carlo

simulations, and derived the effective gauge theory. D.L.-G., Y.I., and M.V. discussed the results, developed the interpretation, and wrote the paper.

## Funding

## Competing interests
The authors declare no competing interests.
