## [Transparent Peer Review File · Nature Communications]

A Classical Chiral Spin Liquid from Chiral Interactions on the Pyrochlore Lattice

Corresponding Author: Dr Daniel Lozano-Gómez

Version 0:

Reviewer comments:

Reviewer #1

(Remarks to the Author)

Lozano-Gomez and co-authors study a theoretical model of classical spins on pyrochlore lattice with three-spin interaction that induces a chirality. They show that ground state of model is equivalent to a four-color model with some special restrictions due to chirality. They claim it is a classical chiral spin liquid. They write a gauge theory for this model and discuss how excitations of this ground state may behave like classical fractons. They also discuss influence of Heisenberg interactions which is the main realistic perturbation of their model. They claim that Heisenberg interaction does not change a priori the nature of ground state.

As cited by the authors in introduction, citation 24 studied a similar model but with biquadratic interaction instead of chirality interaction. The main difference is that model of citation 24 orders and is not a spin liquid. Indeed the cool down simulations done by the authors does not show a phase transition for their model. It suggests maybe spin liquid. But the energy of cool down and warm up simulations are different, and warm up simulations show a peak in specific heat in the supplemental material. This peak is called crossover in supplemental material, but it looks like a phase transition in figure S5. It means that at low temperature, simulations are either frozen in some form of order when warm up, or they are disordered in a phase that is not the ground state when cool down. It is interesting, but not entirely conclusive for the presence of spin liquid.

As cited by the authors in page 6, citation 46 did the field theory with three gauge fields of four-color model on pyrochlore. The authors showed that chirality between spins of equation 1 becomes chirality in field theory in equation 9.

Then the authors explain why a consequence of chirality is that excitations of ground state may be like fractons. This is maybe most interesting result because there is a lot of research on fractons and simple models for fractons are rare. But the authors "leave the study of these charges and a more detailed characterization of the model in Eq. (4) for future work" so it is not sure if there are fractons in simulations.

Maybe fractons are responsible for the excited energy of simulations at low temperature when cool down ?

The paper is easy to read and of interest to people in frustrated magnetism and maybe fracton physics. But some of the claims are not entirely established. For example, are cool down simulations metastable ? Do we really have fractons ? Also, compared to literature, the new results deserve to be published, but I am not sure they have enough impact for Nature Communications.

Small comments

Sometimes the paper is technical. For example, what is a pinch point and bow tie in structure factor ?

A possible paper of interest to the author, Gia-Wei Chern and Congjun Wu, Four-Coloring Model and Frustrated Superfluidity in the Diamond Lattice, PRL 112, 020601 (2014)

(Remarks on code availability)

Reviewer #2

(Remarks to the Author)

In this article, Lozano-Gomez et al. considered chiral interactions between classical spins defined on a pyrochlore lattice and found a classical spin liquid state captured by an effective gauge theory. Starting from the detailed analysis of the chiral interaction defined on a single tetrahedron, they identified 12 distinct ground state configurations. Then, using a classical Monte-Carlo simulation, thermodynamic properties of this model on larger systems were investigated. They further derived an effective 4-state (chiral) Potts model that has a ground state configuration corresponding to that of the original chiral model and developed an effective gauge theory based on an effective Potts model including chiral interactions. The existence of chiral interactions makes the chiral Potts model distinct from the regular Potts model, modifying the ground state manifold and the property of excitations.

I think the authors are successful in describing the emergent gauge structure by chiral interactions, leading to a classical spin liquid state on a 3-dimensional pyrochlore lattice. The paper is written clearly and, therefore, could be published. However, I have a mixed feeling whether or not I should recommend the publication of this work in Nature Communications as explained below.

Introducing chiral interactions on a pyrochlore lattice is a new way to realize a spin liquid state, both ingredients and consequence appear to be novel. However, the behavior of the proposed model does not seem to be so different from the other spin models, such as dipolar interactions on a pyrochlore lattice, showing two-fold pinch points in the spin structure factor and a Coulomb phase of bions. Is there fundamental difference between the different models and between the current classical spin liquid vs. a (quantum) spin ice?

Since a chiral interaction on a triangle requires an (effective) magnetic field penetrating the triangle (Ref. 26), how the interactions are arranged appears to imply that magnetic fluxes are coming from the inside of a tetrahedron to the outside through all triangle surfaces, or vice versa, as if a magnetic monopole is located at the center of the tetrahedron. If I am correct, the physical realization of the current model is extremely difficult, while it may not be entirely impossible. Could this be realized in a physical system, or the current proposal is purely theoretical with no relevance to the real world?

Because of the above points and considering how the paper is written (highly theoretical), the paper may not appeal to the broad audience of Nature Communications. The current manuscript may be better suited to other journal, where people specialized in the frustrated magnetism could enjoy it. Could the authors modify the manuscript so that broad audience could access?

Other suggestions

Some of the subsections may have alternative titles, such as "Regular Potts model" => "Gauge structure of the regular Potts model", "Chiral Potts model" => "Gauge structure of the chiral Potts model", "Chiral and Heisenberg interactions" => "Competition between Chiral and Heisenberg interactions"

(Remarks on code availability)

Version 1:

Reviewer comments:

Reviewer #1

(Remarks to the Author)

I would like to thank the authors for their reply and their efforts. I find this new version clearer, especially related to the emergent physics of fractons, which is, as mentioned in my previous report, probably the most interesting aspect of this paper.

There are still a few questions left unanswered, such as finite-size scaling of the specific heat (and other quantities), or the possibility for an order-by-disorder transition if an efficient non-local spin update were developed in order to thermalise this system at low temperatures (I realise that such an algorithm would be a technical challenge).

But this manuscript has the advantage to present a relatively simple model that sits at the crossing point between chiral spin liquids, fractonic matter and out-of-equilibrium phenomena in a non-disordered magnet. The overall outcome is of sufficient interest that the unanswered questions may actually motivate further studies. It is difficult to say for sure, but this model could become a future toy model for fractons or chiral spin liquidity.

Being easy to read and of interest to an active community of researchers, I would now recommend publication of this improved version of the manuscript.

(Remarks on code availability)

Reviewer #2

(Remarks to the Author)

The authors responded to my questions appropriately and revised the manuscript.

I think the paper is now accessible to the broader audience and, therefore, would like to recommend the publication in Nature Communications.

(Remarks on code availability)

REVIEWER COMMENTS AND RESPONSE

Reviewer #1 (Remarks to the Author):

Lozano-Gomez and co-authors study a theoretical model of classical spins on pyrochlore lattice with three-spin interaction that induces a chirality. They show that ground state of model is equivalent to a four-color model with some special restrictions due to chirality. They claim it is a classical chiral spin liquid. They write a gauge theory for this model and discuss how excitations of this ground state may behave like classical fractons. They also discuss influence of Heisenberg interactions which is the main realistic perturbation of their model. They claim that Heisenberg interaction does not change a priori the nature of ground state.

As cited by the authors in introduction, citation 24 studied a similar model but with biquadratic interaction instead of chirality interaction. The main difference is that model of citation 24 orders and is not a spin liquid. Indeed the cool down simulations done by the authors does not show a phase transition for their model. It suggests maybe spin liquid. But the energy of cool down and warm up simulations are different, and warm up simulations show a peak in specific heat in the supplemental material. This peak is called crossover in supplemental material, but it looks like a phase transition in figure S5. It means that at low temperature, simulations are either frozen in some form of order when warm up, or they are disordered in a phase that is not the ground state when cool down. It is interesting, but not entirely conclusive for the presence of spin liquid.

As cited by the authors in page 6, citation 46 did the field theory with three gauge fields of four-color model on pyrochlore. The authors showed that chirality between spins of equation 1 becomes chirality in field theory in equation 9.

Then the authors explain why a consequence of chirality is that excitations of ground state may be like fractons. This is maybe most interesting result because there is a lot of research on fractons and simple models for fractons are rare. But the authors "leave the study of these charges and a more detailed characterization of the model in Eq. (4) for future work" so it is not sure if there are fractons in simulations. Maybe fractons are responsible for the excited energy of simulations at low temperature when cool down?

We thank the reviewer #1 for his/her overall review of our results and for pointing out the connection between the fractonic excitations and the unusual thermodynamic behavior observed in our simulations.

Indeed, and as the reviewer pointed out, the discrepancy between our warm-up and cool-down simulations is associated with fractonic excitations which are difficult to equilibrate. More specifically, as the temperature is lowered in a cool-down scheme, fractonic excitations tend to freeze in the lattice (due to their restricted mobility) and impede the system from fully accessing its low-energy manifold; this yields a higher internal energy from a cool-down scheme (as opposed to a warm-up scheme). This same behavior has been observed in other fracton systems with concomitant spin-glass-like behavior [R. M. Nandkishore and M. Hermele, *Annu.Rev. Condens. Matter Phys.* 10, 295 (2019)].

To illustrate this point let us consider the two cases we have studied, first the chiral potts model introduced in Eq. (4) of the main text, and then the chiral Hamiltonian in Eq. (1) of the main text, both which are described by the same effective gauge theory in Eq. (9) of the main text.

1. The chiral Potts model

As discussed in the main text, for the chiral potts model introduced in Eq. (4), the ground-state configurations have a vanishing energy, and the elementary excitations are bionic charges and chiral charges. We note that the mobility of these charges is severely restricted, as is discussed and characterized in the main text.

In Fig. 1, we show the internal energy and specific heat per lattice site for both a cool-down and a warm-up scheme, and a single Potts configuration at low temperatures sampled from cool-down classical Monte-Carlo simulations. For the cool-down scheme, our simulations show a smooth evolution of the internal energy, and a Schottky-like peak in the specific heat. We note, however, that the low-temperature internal energy plateaus at a non-vanishing value as $T \rightarrow 0$, implying that excitations are still present in the system. Indeed, by closely inspecting a low-temperature configuration (illustrated in Fig. 1), we see that both bions and chiral excitations are present.

Figure 1. Internal energy per lattice site E , specific heat per lattice site C/k_B , and a single Potts configuration sampled from a classical Monte-Carlo simulation on the chiral Potts model for a system of size $L = 10$. For the Potts configuration the black and purple circles label excited tetrahedra corresponding to a bion and a chiral charge, respectively.

The presence of these charges at low temperatures is associated with their restricted mobility, which in turn lead to a freezing-in of the excitations as the temperature is lowered. A similar picture is seen if we now perform a warm-up scheme for the chiral Potts model starting from a ground-state configuration. In such a case the internal energy sampled by the classical Monte-Carlo does plateau at zero energy at low temperatures. However, and similar to the cool down-scheme, the warm-up scheme is frozen in the initial low-temperature configuration because thermal excitations cannot be easily generated.

2. The chiral Hamiltonian

As in the chiral Potts model, and as noted by the reviewer #1, the Monte-Carlo simulations of the chiral Hamiltonian in Eq. (4) present a similar discrepancy between the energy of the cool-down and warm-up schemes. In the chiral Hamiltonian, however, the bionic and chiral charges are not quantized but are instead continuous degrees of freedom due to the continuous nature of the underlying classical spins. Consequently, thermal spin fluctuations can thermally depopulate these charges. This effect may also be understood as the melting of a charge. A consequence of this thermal depopulation of the charges is that the internal energy of the system sampled from a cool-down scheme slowly decreases with temperature, although it remains above that of the internal energy sampled with a warm-up scheme, as shown in Fig. 7 of the main text. We note that the non-quantized character of the gauge charges for the chiral Hamiltonian refrains us from exactly locating their position in the lattice and producing an analysis similar to the one performed for the chiral Potts Hamiltonian [Flores-Calderon *et al*, arXiv:2402.03083 (2024)]. Instead, we can study the evolution of other statistical quantities which are directly associated with the ground-state manifold. Figure 2 illustrates the nearest-neighbor dot product and the single-triangle chirality for warm-up and cool-down schemes sampled at different temperatures and averaged over 200 configurations. At low temperatures all distributions are centered about the ground-state prediction, however, the distributions for the cool down scheme are consistently

broader than those of the warm up schemes. This difference can be associated with non-vanishing excitations in the system.

Figure 2. Nearest-neighbor dot product (left) and single-triangle chirality (right) for a cool-down scheme on the upper row, and a warm-up scheme on the lower row. The black lines label the expected value of these quantities in an exact ground-state configuration, which is $-1/3$ for the nearest-neighbor dot product and $-4/(3\sqrt{3})$ for the chirality term.

We emphasize that in the warm-up simulations of both models, the system freezes in the initial configuration. Moreover, and as mentioned in our draft, there is an at-least sub-extensive number of ground states for these models which are associated by a cyclic permutation of all spins (or Potts degrees of freedom) in an infinite plane bisecting the pyrochlore lattice, see Fig. 3. We note that all of these ground states are equivalent and as such a warm-up simulation on these would also display the same freezing. In particular, and as shown in section “thermodynamics from warm-up and cool-down schemes” of the Supplement, such an equivalence is shown for the chiral Hamiltonian in Eq. (1) where we show warm-up simulations starting from three distinct ground-state configurations.

Figure 3. Monte-Carlo unit cell with $L=4$ where only the “up” tetrahedra are shown. Here all the blue tetrahedra belong to a single infinite kagome plane which bisects the pyrochlore lattice. The apparent observation of three different planes is a consequence of the periodic boundary conditions of the system.

Let us now comment on how this lack of proper equilibration could be resolved. As we pointed out for both the chiral Potts model and the chiral Hamiltonian, the discrepancy between their temperature-sweeping schemes is associated with remnant charges in the low-temperature phase. To avoid these charges from freezing out in the cool-down scheme and the freezing in a ground-state configuration in the warm-up scheme, a non-local update would be necessary, capable of moving these charges around with no additional energy cost and of connecting different ground-state configurations. In the main text we have discussed the pre-conditions this update must fulfill, however, due to the intricacy of the pyrochlore lattice geometry, we have not been able to implement such a non-local move in our simulations.

Indeed, a recent work on a classical spin liquid with fractonic excitations showed that such an algorithm can radically improve the thermodynamic behavior of the simulation [B. Placke *et al.*, arXiv:2306.13151 (2023)]. In that case, however, the fractonic excitations they identify are partially mobile which allows for the construction of such moves. This is not the case for the models we considered and therefore complicates the full identification and construction of such non-local updates.

To summarize, we have studied two systems which are described by the same effective gauge theory at low temperatures. These systems possess fracton excitations with restricted mobility where the excitations are bions and chiral charges. In a cool down Monte-Carlo simulation, the restricted mobility of these charges results in freezing at low temperatures in both systems. The presence of these excitations is exposed by the study of the configuration in the Potts model and the distribution of the nearest-neighbor dot product and chirality term for the chiral Hamiltonian. Additionally, in a warm up Monte-Carlo simulation starting from a ground-state configuration, the restricted mobility results in the freezing of the system in the initial configuration at low temperatures. To overcome the discrepancy between the different temperature schemes, a non-local algorithm would be necessary. In the main text of our draft we have outlined what are the conditions such a non-local update must fulfill and showed a family of these non-local updates corresponds to the permutation of all spins (Potts degrees of freedom) in an infinite plane bisecting the system.

In our efforts to clarify this point in our draft we have added the following text to the main text,

“The restricted motion of these excitations results in glassy dynamics typically observed in fracton systems~\cite{Hermele_fractons_2019}. Indeed, performing a warm-up and cool-down classical Monte-Carlo

simulation on the chiral Potts model exposes a similar behavior to the one observed in Fig.~\ref{fig:MC_warm_up_cool_down}: in the cool-down scheme, the restricted motion results in freezing of charges at low temperatures yielding a higher internal energy in these simulations. In the warm-up scheme, the restricted motion results in freezing of the simulations at low temperatures about the initial configuration, we refer the reader to the SI for further details on the chiral Potts model. Similarly, for the chiral model in Eq.~\eqref{eq:chiral_Hamiltonian}, the fractonic nature of these charges is responsible for the disagreement between the internal energy illustrated in Fig.~\ref{fig:MC_warm_up_cool_down}, as well as the histograms of the nearest-neighbor spin correlation in Fig.~\ref{fig:MC_dot_product_warm_cool} measured in the warm-up and cool-down schemes. In the chiral Hamiltonian, however, the continuous nature of the spin degrees of freedom allows for a slow thermal depopulation of the gauge charges and therefore a decreasing internal energy with decreasing temperature. The presence of these charges however is reflected in the width of the histograms in Fig.~\ref{fig:MC_dot_product_warm_cool} which are consistently broader in the cool-down simulations. For more details regarding the difference between the cool-down and warm-up schemes for the chiral Hamiltonian in Eq.~\eqref{eq:chiral_Hamiltonian} we refer the reader to the SI.”

In addition, we have modified two sections in the supplement. First, we have modified the section “Effective Potts Model” to include the warm-up Monte-Carlo for the Potts model shown in Fig. 1, along with the following text

“A similar picture is seen when performing a warm-up scheme starting from a ground-state configuration, see red curve in Fig.~\ref{fig:Potts_model}(a) and (b). In such a case the internal energy sampled by the classical Monte-Carlo does plateau at zero energy at low temperatures. However, and similar to the cool-down scheme, the warm-up scheme is frozen in the initial low-temperature configuration due to the high cost associated with generating and moving the fractonic charges.”

and the section “Thermodynamics from warm-up and cool-down schemes” by including the chirality distributions shown in Fig. 2 along with the following text

“... The peak in the specific heat observed in Fig.~\ref{fig:MC_warm_up_cool_down_E_and_C} in the warm-up simulations is associated with the freezing of the system into the initial partially ordered configuration. Indeed, our results suggest that the freezing observed in the warm-up scheme is bound to take place irrespective of the initial ground-state configuration selected.

Similar to the chiral Potts model, the discrepancy between the warm-up and cool-down schemes can be associated with the presence of non-vanishing charges. In the chiral Hamiltonian, however, the bionic and chiral charges are not quantized but are instead continuous degrees of freedom which can be thermally depopulated by spin fluctuations. A consequence of such a thermal depopulation is that the internal energy of the system sampled from a cool-down scheme slowly decreases with temperature, although it remains above that of the internal energy sampled with a warm-up scheme, as shown in Fig.~\ref{fig:MC_warm_up_cool_down} of the main text. Furthermore, the non-quantize character of the gauge charges for the chiral Hamiltonian refrains us from exactly locating their position in the lattice and producing an analysis similar to the one performed for the chiral Potts Hamiltonian in Fig.~\ref{fig:Potts_model}~\cite{florescalderon2024irrational}. Instead, we can study the evolution of other statistical quantities which are directly associated with the ground-state manifold. Figure~\ref{fig:MC_dot_product_warm_cool} in the main text and Figure~\ref{fig:MC_chirality_warm_cool} illustrates the nearest-neighbor dot product and the single triangle chirality for warm-up and cool-down schemes sampled at different temperatures and averaged over 200 configurations, respectively. At low temperatures, all distributions are centered about the ground-state prediction (this being $-1/3$ for the nearest-neighbor dot product and $-4/(3\sqrt{3})$ for the chiral χ_{ijk} term). However, the distributions for the cool-down scheme are consistently broader than those of the warm-up schemes. This difference is associated with non-vanishing gauge charge excitations in the system.

Lastly, we restate that a further study of both the chiral Hamiltonian in Eq.~\eqref{eq:chiral_Hamiltonian} and the chiral Potts Hamiltonian in Eq.~\eqref{eq:Potts_Hamiltonian} necessitates the implementation of a

non-local update~\cite{placke2023ising} capable of avoiding the freezing observed in the simulations and characterized in the main text. However, we emphasize that the analysis performed in this work already characterizes the intricate and rich physics which are observed in this system.”

The paper is easy to read and of interest to people in frustrated magnetism and maybe fracton physics. But some of the claims are not entirely established. For example, are cool down simulations metastable ? Do we really have fractons ? Also, compared to literature, the new results deserve to be published, but I am not sure they have enough impact for Nature Communications.

We thank the referee for his/her careful assessment of our work while pointing out the possible confusion on the fracton excitations. We have addressed this criticism above and modified the main text and supplementary information accordingly.

We continue to believe that our work is of great interest to both the frustrated-magnetism and fracton communities. In particular, our work sits at the junction of these two fields and also represents a unique case where fracton excitations can be realized in a simple model with realistic spin exchange interactions. The novelty of our findings is two-fold, (i) it establishes an unforeseen and highly nontrivial connection between time-reversal symmetry breaking and fractons. It thus puts forth a novel mechanism for realizing fractons in a whole new class of spin models, a problem which is much in vogue, and (ii) it establishes a whole new *genre* of spin liquids by expanding to the classical regime, the paradigm of Kalmeyer-Laughlin chiral quantum spin liquids about three decades after they were introduced. The Kalmeyer-Laughlin chiral spin liquid (and the parent fractional quantum Hall state) introduced the notion of topological order and opened an entire field of research with classifying topological and quantum orders *à la* X.-G. Wen. In similar spirit, our model of a classical chiral spin liquid in the classical domain, along with its effective theory, opens exciting new avenues for further research. It calls for an exploration and development of a mathematical classification scheme and formalism which would enable distinguishing different types of classical chiral spin liquids and the associated orders. Thus, in addition to its impact, given the profoundness and broad character of the results, our work will be of appeal and kindle the interest of a large community of researchers.

Small comments

Sometimes the paper is technical. For example, what is a pinch point and bow tie in structure factor ?

A possible paper of interest to the author, Gia-Wei Chern and Congjun Wu, Four-Coloring Model and Frustrated Superfluidity in the Diamond Lattice, PRL 112, 020601 (2014)

We thank the referee for pointing out this technical detail which may result in a confusion for a general reader. We have now added an explanation as follows:

“As the temperature is decreased these features sharpen up leading to the observation of two-fold pinch points~\cite{Isakov-2004,Pretko_fractonsPhysRevB.98.115134,Pretko-2017,yan2023classification_1,yan2023classification_2,Castelnovo-2012,Benton_topological_PhysRevLett.127.107202,Davier_spanish_group_PhysRevB.108.054408,Chern2014fourColoring}, see Fig.~\ref{fig:Sq}(b), resulting in connected bow tie and diamond patterns in the $[\text{hh}\ell]$ and $[\text{hk}0]$ planes, respectively. The two-fold pinch point features reflect dipolar correlations between the spin degrees of freedom and are indicative of an energetically imposed Gauss' law constraint on certain gauge field $\mathbf{B}^{\text{(c)}}$, namely $\nabla \cdot \mathbf{B}^{\text{(c)}}=0$, describing an effective low-temperature theory of the system~\cite{Chern2014fourColoring,Moessner-Chalker-98}.”

Reviewer #2 (Remarks to the Author):

In this article, Lozano-Gomez et al. considered chiral interactions between classical spins defined on a pyrochlore lattice and found a classical spin liquid state captured by an effective gauge theory. Starting from the detailed analysis of the chiral interaction defined on a single tetrahedron, they identified 12 distinct ground state configurations. Then, using a classical Monte-Carlo simulation, thermodynamic properties of this model on larger systems were

investigated. They further derived an effective 4-state (chiral) Potts model that has a ground state configuration corresponding to that of the original chiral model and developed an effective gauge theory based on an effective Potts model including chiral interactions. The existence of chiral interactions makes the chiral Potts model distinct from the regular Potts model, modifying the ground state manifold and the property of excitations.

I think the authors are successful in describing the emergent gauge structure by chiral interactions, leading to a classical spin liquid state on a 3-dimensional pyrochlore lattice. The paper is written clearly and, therefore, could be published. However, I have a mixed feeling whether or not I should recommend the publication of this work in Nature Communications as explained below.

Introducing chiral interactions on a pyrochlore lattice is a new way to realize a spin liquid state, both ingredients and consequence appear to be novel. However, the behavior of the proposed model does not seem to be so different from the other spin models, such as dipolar interactions on a pyrochlore lattice, showing two-fold pinch points in the spin structure factor and a Coulomb phase of bions. Is there fundamental difference between the different models and between the current classical spin liquid vs. a (quantum) spin ice?

We thank the reviewer #2 for raising the important question on how “our” chiral spin liquid can be fundamentally differentiated from other spin liquids that have been previously realized in the pyrochlore lattice. As the reviewer pointed out, and not unlike in classical spin ice, the ground-state manifold of the novel chiral spin liquid is characterized by absence of bionic charges and the low-energy description based on divergence-free gauge fields, resulting in the observation of two-fold pinch points in the correlation functions. The chiral spin liquid, however, also possesses an additional constraint on the gauge fields which restricts the chirality of the total gauge flux in every tetrahedra; such a constraint is introduced by the chiral interaction in the Hamiltonian and is described by the second term in Eq. (9). Although this additional constraint seems rather harmless and inconsequential, it has severe ramifications in the gauge fields and its associated charges in the system. Indeed, and as pointed out in section “Excitations of the chiral Potts model” of our draft, this constraint identifies an additional excitation associated with having a left-hand chirality in the total gauge flux of a single tetrahedra even when no bionic charge is present. The inclusion of this charge results in the confinement of the bionic charges, which in the regular Potts model (and similar to the monopoles in classical spin ice) are free to move. Additionally, in section “Excitations of the chiral Potts model”, we comment on how this new chiral constraint severely restricts the motion of both of these charges, which is one of the tell-tale signs we use to identify these as fractonic charges. Altogether, the inclusion of the left-hand chiral charge and its restricted mobility leads to a spin-liquid phase whose gauge charges are fundamentally different from those of the Coulomb spin liquids mentioned by the reviewer.

To emphasize this point and to echo the fundamental distinction between this novel chiral spin liquid and formerly introduced spin liquids in the pyrochlore lattice, we have modified the following text in the discussion of our results.

“We emphasize that the restriction on the chirality of the gauge fluxes and the restricted motion of the associated gauge charges fundamentally differentiate the chiral spin liquid realized for the model in Eq. (1) and (10) from previously identified spin liquid phases in the pyrochlore lattice. Additionally, and of particular interest for the study of fracton models, the realization of fractonic charges identifies the chiral Hamiltonian in Eq. $\sim \text{Eq. (9)}$ on the pyrochlore lattice as a “simple” fracton model whose further study may shed light on the intricate physics associated with these systems.”

Since a chiral interaction on a triangle requires an (effective) magnetic field penetrating the triangle (Ref. 26), how the interactions are arranged appears to imply that magnetic fluxes are coming from the inside of a tetrahedron to the outside through all triangle surfaces, or vice versa, as if a magnetic monopole is located at the center of the tetrahedron. If I am correct, the physical realization of the current model is extremely difficult, while it may not be entirely impossible. Could this be realized in a physical system, or the current proposal is purely theoretical with no relevance to the real world?

As the reviewer #2 pointed out, the chiral interaction we considered is partially motivated by its derivation on Ref. 26 of our paper, which derives the chiral interaction in a t/U expansion when an external magnetic field is applied. For

the pyrochlore lattice, this interaction would indeed require a *local* magnetic field which points from every sublattice position into the center of every single up (or down) tetrahedron. Such a field configuration cannot be produced externally, however, such a field is known to occur internally in the family of the pyrochlore iridates $A_2Ir_2O_7$, with A a rare-earth ion. In this family of compounds, the Ir ions may undergo a phase transition into an all-in-all-out symmetry-breaking phase at elevated temperatures compared to the strength of the exchange interaction typically observed for the rare-earth ions in the A sites [Witczak-Krempa *et al*, Annu. Rev. Condens. Matter Phys. 5, 57 2014]. The magnetic order in the Ir ions results in an effective molecular field along the local z direction for the A sites. The manifestation of a weak local magnetic field on rare-earth ions on the A sites consequently identifies the family $A_2Ir_2O_7$ compounds to be the natural candidates for the realization of the chiral spin liquid we have introduced and studied in our work.

Because of the above points and considering how the paper is written (highly theoretical), the paper may not appeal to the broad audience of Nature Communications. The current manuscript may be better suited to other journal, where people specialized in the frustrated magnetism could enjoy it. Could the authors modify the manuscript so that broad audience could access?

We thank the reviewer #2 for pointing out how the current presentation of our work may be missing an important connection to the experimental realization of the classical spin liquid we have uncovered. Based on our previous comment on the realization of this spin Hamiltonian in the pyrochlore iridates $A_2Ir_2O_7$, as well as the possible observation of associated thermodynamic features for systems in the proximity of this model, we have included the following text in the main body of our work to point out the possible candidate compounds to realize this novel chiral spin liquid.

“As we have previously discussed, the chiral interaction in Eq.(1) descends from a t/U expansion where an external magnetic field is applied [ref. 26]. The realization of such an exchange term in the pyrochlore lattice would imply the presence of a *local* magnetic field pointing towards (away) the center of each tetrahedra. Such a local magnetic field is known to take place in the family of the pyrochlore iridates, described by the chemical formula $A_2Ir_2O_7$, where the A ions are typically rare earth elements, and both the A and the Ir ions occupy two interpenetrating pyrochlore lattices. In this family of compounds, the Ir ions may undergo a phase transition into an all-in-all-out symmetry-breaking phase at relatively high-temperatures compared to the strength of the exchange interaction typically observed for the rare-earth ions in the A sites [Witczak-Krempa *et al*, Annu. Rev. Condens. Matter Phys. 5, 57 2014]. The magnetic order in the Ir ions result in a relative weak effective molecular field along the local z direction for the rare-earth ions on the A sites [Witczak-Krempa *et al*, Annu. Rev. Condens. Matter Phys. 5, 57 2014]. Following Ref.[26], the introduction of a weak local magnetic field on rare-earth ions on the A sites may result in the chiral spin interactions we considered. Consequently, we identify the family $A_2Ir_2O_7$ of compounds where the Ir ions order into an all-in-all-out symmetry breaking phase to be the natural candidates for the realization of the chiral interaction and therefore the chiral spin liquid we have introduced and studied in the present work. “

Furthermore, we note that even if the Hamiltonian for a candidate material in this family of compounds does not exactly match the interaction couplings considered in Eq. (10), the sole proximity to this spin liquid may yield remnant thermodynamic features associated with the spin liquid. Indeed, such remnant features behavior have been predicted and observed for other spin liquids in the pyrochlore lattice, this being the case for $Yb_2Ti_2O_7$ [A. Scheie *et al*, Phys. Rev. Lett. 129, 217202 (2022)] and FeF_3 [A. Sadeghi *et al*, Phys. Rev. B 91, 140407 (2015)], and even for other frustrated lattices as is the case for the recently synthesized trillium lattice compounds $K_2Ni_2(SO_4)_3$ [M. G. Gonzalez *et al*, Nature Communications 15, 7191 (2024)].”

Other suggestions

Some of the subsections may have alternative titles, such as “Regular Potts model” => “Gauge structure of the regular Potts model”, “Chiral Potts model” => “Gauge structure of the chiral Potts model”, “Chiral and Heisenberg interactions” => “Competition between Chiral and Heisenberg interactions”

We thank the reviewer #2 for suggesting the alternative titles for the sections we have proposed in our draft. Accordingly, we have decided to update two of the three suggested titles, which are now implemented in the main text.

“Regular Potts model” => “Gauge structure of the regular Potts model”

“Chiral Potts model” => “Gauge structure of the chiral Potts model”

List of edits performed to the submitted draft:

In the main text:

1. Edits to “Results” section in the subsections “numerical results”, “Excitations of the chiral Potts model”.
2. Edits to the “Discussion” section.
3. Change to the title of subsections previously named “Regular Potts model” and “Chiral Potts model”.

In the Supplementary Information:

1. Edits to the “Effective Potts model” and “Thermodynamics from warm-up and cool-down schemes” sections.
2. Additional Figure S.7.

REVIEWER COMMENTS AND RESPONSE

Reviewer #1 (Remarks to the Author):

I would like to thank the authors for their reply and their efforts. I find this new version clearer, especially related to the emergent physics of fractons, which is, as mentioned in my previous report, probably the most interesting aspect of this paper.

There are still a few questions left unanswered, such as finite-size scaling of the specific heat (and other quantities), or the possibility for an order-by-disorder transition if an efficient non-local spin update were developed in order to thermalise this system at low temperatures (I realise that such an algorithm would be a technical challenge).

But this manuscript has the advantage to present a relatively simple model that sits at the crossing point between chiral spin liquids, fractonic matter and out-of-equilibrium phenomena in a non-disordered magnet. The overall outcome is of sufficient interest that the unanswered questions may actually motivate further studies. It is difficult to say for sure, but this model could become a future toy model for fractons or chiral spin liquidity.

Being easy to read and of interest to an active community of researchers, I would now recommend publication of this improved version of the manuscript.

We would like to thank reviewer #1 for his positive assessment of our revised manuscript, highlighting its long-term impact, and his/her recommendation for the publication of our work in Nature Communications. We agree with the reviewer regarding the importance of a finite size scaling illustrating the preservation of the classical chiral spin liquid in the thermodynamic limit. To address this remark we have included an additional appendix to the supplementary information stating the following

“To ensure that the analysis that we have presented is valid in the thermodynamic limit, we have performed a cMC simulations of the chiral Hamiltonian in Eq.~\eqref{eq:chiral_Hamiltonian} for three distinct system sizes with $N=4L^3$ spins, namely $L=6, 8, 10$. The specific heat obtained for these systems is illustrated in Fig.~\ref{fig:MC_chirality_finite_size} where the double bump feature in the specific heat is preserved for all the systems studied. The agreement of the specific heat between these three system sizes suggests that the classical chiral spin liquid is indeed the phase realized by the chiral Hamiltonian in the thermodynamic limit at low temperatures.”

Supplementary Figure 9. Specific heat for the chiral Hamiltonian in Eq.~\eqref{eq:chiral_Hamiltonian} sampled through classical Monte-Carlo for three system sizes, namely $L=6,8,10$.

Reviewer #2 (Remarks to the Author):

The authors responded to my questions appropriately and revised the manuscript. I think the paper is now accessible to the broader audience and, therefore, would like to recommend the publication in Nature Communications.

We would like to thank reviewer #2 for his positive assessment of our work and his recommendation for publication.